# Agreement ℜ of Four Analytical Methods Applied to Pb in Soils from the Small City of St. John’s, Newfoundland, Canada

**DOI:** 10.3390/ijerph18189863

**Published:** 2021-09-18

**Authors:** Christopher R. Gonzales, Anna A. Paltseva, Trevor Bell, Eric T. Powell, Howard W. Mielke

**Affiliations:** 1Department of Pharmacology, Tulane University School of Medicine, 1430 Tulane Ave., New Orleans, LA 70112, USA; chrisgc99@gmail.com; 2School of Geosciences, University of Louisiana,104 East University Avenue, Lafayette, LA 70504, USA; anna.paltseva@louisiana.edu; 3Department of Landscape Design and Sustainable Ecosystems, Agrarian-Technological Institute, RUDN University, Miklukho-Maklaya Street, 6, 117198 Moscow, Russia; 4Department of Geography, Memorial University of Newfoundland, 230 Elizabeth Ave, St. John’s, NL A1C 5S7, Canada; tbell@mun.ca; 5Lead Lab, Inc., 3233 DeSoto, New Orleans, LA 70119, USA; powellet2@gmail.com

**Keywords:** urban soil, soil extractions, bioaccessibility, Berry–Mielke’s Universal ℜ agreement statistics, human exposure, pXRF, blood lead

## Abstract

In the small city of St. John’s, NL (2020 population ~114,000), 100% of the soils of the pre-1926 properties exceeded the Canadian soil Pb standard, 140 mg/kg. The Pb was traced to high-Pb coal ash used for heating and disposed on the soils outside. Analytical instruments became available in the late 1960s and 1970s and were first used for blood Pb and clinical studies and repurposed for measuring environmental Pb. The environmental research part of this study compared four common soil Pb analysis methods on the same set (N = 96) of St. John’s soil samples. The methods: The US EPA method 3050B, portable X-ray fluorescence spectrometry (pXRF), The Chaney–Mielke leachate extraction (1 M nitric acid), and the relative bioaccessibility leaching procedure (US EPA method 1340). Correlation is not the same as agreement ℜ. There is strong agreement (Berry–Mielke’s Universal ℜ) among the four soil Pb analytical methods. Accordingly, precaution is normally advisable to protect children from the high-Pb garden soils and play areas. A public health reality check by Health Canada surveillance of St. John’s children (N = 257) noted remarkably low blood Pb. The low blood Pb of St. John’s’ children is contrary to the soil Pb results. Known urban processes causing the rise of environmental Pb and children’s Pb exposure includes particle size, aerosol emission by traffic congestion, and quantities of leaded petrol during the 20th century. Smaller cities had minor traffic congestion and limited combustion particles from leaded petrol. From the perspective of the 20th century era of urban Pb pollution, St. John’s, NL, children have blood Pb characteristics of a small city.

## 1. Introduction

During the 1960s, instruments became available for analyzing small quantities of physiologically relevant Pb [1]. The atomic absorption spectrometer (AAS) was applied to monitoring children’s blood Pb and clinical outcomes of Pb. Increasing clinical awareness about health damage from Pb exposure became known, and the response was a stepwise reduction of blood Pb guidelines. Children’s blood Pb assisted in finding sources of environmental Pb, a secondary prevention practice because exposure sources were found after children are exposed. The same instruments were repurposed as environmental research tools for the monitoring of Pb in air, water, and soil media, and analytical instruments became available as tools for primary prevention to avert children’s exposure in the first place. 

This study was instigated when student researchers discovered elevated soil Pb on residential properties in the small city of St. John’s, Newfoundland and Labrador (NL). Environmental monitoring of soil Pb was conducted at Memorial University of Newfoundland [2]. The soil in St. John’s (2020 population ~114,000) had Pb similar to other Canadian cities, including communities polluted by industrial Pb. In St. John’s, almost all of the pre-1926 properties exceeded the 140 mg/kg Canadian soil Pb guideline. The study also indicated that the geometric mean soil Pb concentration is appreciably different for homes built before 1970 (187 mg/kg), compared to homes built post-1980 (28.5 mg/kg). Soil Pb in the older parts of St. John’s originated from high-Pb coal used for heating. Over decades, paint flaked from wooden clapboard houses and coal ashes were spread outdoors on residential properties. Across all housing ages, soil along the drip line has higher median Pb concentration (194 mg/kg) than the median soil Pb reported (in mg/kg) for ambient (138, N = 514)), and road samples (136, N = 389). The median for all samples combined (N = 1231) was 148 mg/kg [2]. 

Soil samples for this analysis were collected and prepared by a research team from Memorial University of Newfoundland (MUN) in St. John’s. The soil samples were analyzed with US EPA Method 3050B [3]. The procedure uses high temperatures and concentrated acids to extract metals from soils and analyzed by inductively coupled plasma-mass spectrometry (ICP-MS). After analysis with US EPA method 3050B, we collaborated with Dr. Bell to obtain the samples and compare their results with other analytical methods. The soil samples were shipped to Tulane University School of Medicine, New Orleans, for analysis by three other methods.

The aim was to compare the agreement of four analytical methods for Pb on the same set of soil samples (N = 96). The soil Pb analysis methods cover a spectrum of commonly used soil Pb analytical procedures. We sought to compare the results of the methods to find out how they agree with each other. The comparison investigation took on special meaning because, after decades of studies with the Chaney–Mielke method (developed in the mid-1970′s), this would fill a needed perspective on results obtained by studies using these analytical methods. The following paragraph describes the other three additional Pb analysis methods.

The Chaney–Mielke method uses a 1 M nitric acid leachate for the extraction of metals from soils [4]. The method was created to provide a relatively low cost, safe, room-temperature analysis of Pb and other metals in soil, and it was first applied to map metals in Baltimore’s garden soils [5]. The relative bioaccessibility leaching procedure, or U.S. EPA method 1340, was created to mimic soil Pb bioaccessibility at gastric pH and temperature [6]. Portable energy dispersive X-ray fluorescence spectrometry (or pXRF) is non-destructive and does not involve extraction. Analysis of the St. John’s soil samples (N = 96) was also conducted with a portable Niton XL2 XRF analyzer (Niton XL2 XRF Analyzer, Thermo Fisher Scientific, Waltham, MA, USA) [7,8]. 

Correlation statistics are often used for method comparison. This study distinguishes between agreement and correlation statistics. The tests for agreement were calculated for the results obtained by the four methods. In practice, agreement statistics are the correct test for comparisons of measurements of the same variable by different methods. For this study, agreement was evaluated with the Berry–Mielke’s Universal ℜ test [9]. 

As a check of the effect of soil Pb on blood Pb, the MUN research team conducted a survey of the children of St. John’s funded by Health Canada [10]. Public health concerns about the potential Pb exposure and health of St. John’s children were raised because soil Pb studies indicated the possibility of excessive Pb exposure. Children are especially vulnerable to Pb exposure, and the effects are particularly egregious [11,12]. For children, soil Pb presents multiple pathways of exposure, including hand-to-mouth ingestion of soil Pb, inhalation of air Pb, inhalation of Pb dust resuspended from Pb contaminated soil, and the track-in of Pb dust into homes from contaminated soils [13,14]. The U.S. Centers for Disease Control and Prevention (CDC) advises that there is *no known safe level of Pb* [15]. The essential need is primary prevention and intervention programs before children are Pb-exposed [16]. 

The four methods selected for this study are analytical procedures generally available to students and researchers. We describe agreement results, list issues for each analytical method, provide soil Pb and children’s blood Pb results, and suggest factors that play an essential role in children’s exposure from soil Pb. The goal is to understand urban conditions where environmental Pb is a root cause of concern. 

## 2. Materials and Methods

The development of analytical methods includes decisions about measuring the relevant Pb in soil. For example, concentrated acids, high temperature, and elevated pressure are employed to extract as much metal as possible. Some methods use less concentrated acids or leachates to mimic the human stomach or attempt to match the method with physiological processes of the digestive tract. 

### 2.1. Soil Collection and Initial Preparation

The Geological Survey of Canada (GSC) determined the background soil Pb concentrations in glacial till throughout Canada for the particle size fraction < 63 μm; Pb concentration ranged from 1 to 152 mg/kg, with a median of 8 mg/kg and a 90th percentile of 16 mg/kg (N = 7398) [17]. The <63 µm particle size was deemed significant because house dust and soil particles of these sizes adhere to children’s hands [18,19]. Moreover, this particle size is more likely to dissolve in the stomach and travel across the gastric mucosa [20]. 

The St. John’s soil samples (96 of 421 collected and analyzed) were part of a MUN “LeadNL” project. Surficial soil samples from the top 0–10 cm were collected from properties of St. John’s, NL (Bell 2021, personal communication). The samples were collected from residential yards and vegetable gardens. There were roughly twice the number of soil samples collected from the older section of St. John’s with pre-1970 housing and from communities with housing built post-1980. The GSC protocol was used, and the soils were dried overnight at 40 °C, sieved to <63 µm size with stainless steel sieves, and stored in low-density polyethylene bags.

### 2.2. Method Comparisons on St. John’s Contaminated Soil 

There are many Pb assessment methods used in the USA and throughout the world [21]. Advanced studies such as a Pb isotope measurements study in London determined the links between current atmospheric Pb and leaded petrol used during the 20th century [22]. Moreover, a Pb isotope study was conducted to determine the anthropogenic sources of Pb deposited in the topsoils from rural areas in the Netherlands [23]. Four common methods were selected to compare agreement of the soil Pb analyses results. All analytical methods were applied to the same soil samples.

(1)The US EPA method 3050B was performed at Maxxam Analytics in Bedford, Nova Scotia, an accredited ISO 17,025 certified laboratory that conducts analysis with inductively coupled plasma-mass spectrometry (ICP-MS). The detection limit for Pb in soil is 0.5 mg/kg [3]. The method is not a total metal extraction technique because elements bound in silicate structures are not normally dissolved by this procedure [3]. Then, after analysis with the US EPA Method 3050B, the results, along with the dried and sieved St. John’s NL soil samples (N = 96), were shipped to Tulane University in New Orleans to conduct Pb analysis using the following three Pb analysis methods:(2)Portable energy dispersive X-ray fluorescence spectrometry (pXRF). Analysis was conducted with a Niton XL2 handheld XRF for 30 s on <63 µm particle size soil samples. NIST standards and a common laboratory standard were used to check the calibration. The detection limit was 15 mg/kg [7,8,24].(3)Chaney–Mielke extraction (1 M nitric acid) is a room temperature leachate extraction consisting of 400 mg of <63 µm particle size sample, mixed with 20 mL 1 mol/L nitric acid for 2 h on a shaker, followed by centrifuging, filtering, and analysis with an inductively coupled plasma-optical emission spectrometer (ICP-OES). Detection limit for Pb is 2 mg/kg [25].(4)The relative bioaccessibility leaching procedure (RBALP) is also known as US EPA method 1340 [6]. One gram of <63 µm soil sample was added with 100 mL of 0.4 M glycine buffer, pH 1.5, preheated to 37 °C, in a 125 mL HDPE bottle and placed into a rotating shaker located in a 37 °C incubator for 1 h at 30 ± 2 rpm speed. The solution pH was frequently checked and adjusted with HCL to 1.5 ± 0.05. After 1 h, an aliquot of suspension was collected and filtered through 0.45 μm filters followed by ICP-OES measurement. The detection limit for Pb is 2 mg/kg [26].

### 2.3. Measurements of Agreement

Agreement refers to the degree of consistency, concordance, or reliability between two or more sets of measurements of the same variable (in this case Pb). Agreement is not the same as correlation. Correlation refers to whether two variables show a linear relationship. The results for each analytical method were evaluated for agreement by calculating the Berry–Mielke Universal ℜ coefficient of agreement [27]. The ℜ coefficient is a generalization of Cohen’s kappa to an interval and ordinal measurement scale. It allows for the data comparison of results from two or multiple methods. The coefficient is chance-corrected and appropriate for the measurement of reliability between results. It is based on Euclidean distances in a multivariate framework, and its significance is tested using Pearson Type III distribution with StatsDirect St V 3.2.10. statistical software (StatsDirect, Ltd., Merseyside, UK) [28]. 

In addition to agreement, as illustrated in the Appendix A, the methods were evaluated using the least absolute deviation regression (LAD), which minimizes the absolute value of the residuals and provides a robust solution when outliers are present [29]. LAD estimates the conditional median of the response variable. 

## 3. Results

### Comparison of the Four Soil Pb Analysis Methods

Table 1 shows the results for each method as percentiles, and comparisons of Universal ℜ coefficients of agreement. Note the small *p*-values (<0.0001) for the comparisons. The EPA 3050B method resulted in Pb concentrations ranging from 13 to 1900 mg/kg, while the pXRF measured higher concentrations that were between 15 and 2058 mg/kg. Soil Pb extractions with 1 M nitric acid resulted in a range from 10 to 1410 mg/kg, and the RBALP method extracted less Pb than the other three methods (7 to 1295 mg/kg of Pb).

Table 1 summarizes the descriptive statistics (percentiles) on the left, and the agreement ℜ results are on the right. Along with the Berry–Mielke ℜ results for the four methods, the *p*-values are also shown. Comparing methods, the pXRF vs. EPA 3050B and the 1 M nitric acid vs. EPA 3050B, the Universal ℜ test showed the strongest coefficients of agreement of 0.832 and 0.833, respectively. The agreements demonstrate the value of the Chaney–Mielke extraction method for the analysis of soil Pb. The Appendix A illustrate six pairwise graphs of the soil Pb analysis methods. Each graph shows the 45° line of equality and the regression equation with the coefficient of determination (R^1^). Although there is a strong correlation between methods this does not indicate agreement between the methods. 

The median soil Pb results shown in Table 1 are between 113 and 59 mg/kg, depending on the method. The differences in percentiles on the left in Table 1 are not indicative of agreement or disagreement, as shown by the Berry–Mielke ℜ results on the right. The percentiles do not correspond to the same soil sample. Figure 1 illustrates the data for each soil sample analyzed individually for each method and ranked by pXRF value. Note the spread of the results for the most Pb-contaminated soil samples. The strongest agreements (ℜ = 0.765–0.833) were among the EPA 3050B, 1 M nitric acid, and pXRF methods. The weakest agreement (ℜ = 0.552) was between the pXRF and RBALP methods. The Universal ℜ test of agreement was also calculated for the four methods and shows strong agreement (ℜ = 0.727, *p*-value < 0.0001). 

## 4. Discussion

Anthropogenic mining, smelting, manufacturing, and commercial activities have redistributed massive quantities of industrial lead (Pb) into the Earth’s total ecology [30,31]. First, conducting measurements of anthropogenic Pb uncovers important practical matters. Secondly, we review the LeadNL’s findings from a children’s blood Pb survey in St. John’s. Thirdly, the LeadNL results compel us to reconsider research methodology and the environmental characteristics of St. John’s, a small non-industrial city with minimal traffic flows and automobile congestion. 

### 4.1. Practical Matters concerning Soil Pb Analysis Methods on soils St. John’s NL 

The experience of conducting Pb analysis has given our study team the opportunity to compare the practical use of each of the methods. Urban environmental surveying of soil Pb involves many issues, including the relevance of the measurements to health outcomes, safety concerns, and ease of measurement. Other characteristics include cost of the extraction equipment, cost of the analytical equipment, time and effort required to prepare soil samples, and productivity for each method. These characteristics are listed in Table 2. 

As shown in Table 2, each method has strengths and weaknesses. The EPA 3050B method as used here was designed to extract multiple metals in soil at high temperature and pressure. The method uses concentrated acid digestion that will dissolve almost all metals that could become environmentally available. It is equipment-intensive and requires special hoods and safety training by personnel to perform the microwave extractions. The laboratory and personnel costs are relatively large, especially for the extraction and operation of the ICP-MS, and the sample throughput per day is relatively limited. 

The 1 M nitric acid method uses room-temperature sample preparation and is relatively safe to perform. The method requires a simple shaker apparatus, no hoods, and it is suitable for conducting metal analysis with a low cost AAS instrument. After extraction and filtering, up to a hundred samples can be analyzed per day. Also, based on community-based surveys of urban soil Pb matched with surveys of children’s blood Pb in New Orleans, the method has been proven relevant because soil Pb is strongly associated with children’s blood Pb [14,32].

RBALP is a soil extraction method assumed to estimate bioaccessible Pb. This method requires a constant temperature and pH adjustments to mimic the stomach during soil metal extraction. It is a time-consuming and laborious procedure, and this limits the number of samples processed per day. Traditional laboratory methods with advanced equipment are often costly. A simpler bioaccessible method is to apply the pXRF analyzers for measuring both total metal concentrations in solids and the extractable metal concentrations in liquids (e.g., extraction with glycine-HCl) [33,34]. A less complicated method for measuring potential oral bioavailability is the use of 0.43 1 M nitric acid, and the results correlate with the RBALP method results [35].

The easiest to use soil metal measuring method is the pXRF. It reliably quantifies metals in fresh, bulk samples in the laboratory. Especially importantly, it can be used directly in the field. In this study, laboratory pXRF measurements had high agreement and correlated well with the other methods. The R^1^ for the LAD ranged from 0.789 to 0.841, and agreement ℜ ranged from 0.552 to 0.832 *p*-values < 0.0001. Other researchers have reported that pXRF is reliable when precision is within 20%, with target elements’ concentrations > 10 times the pXRF detection limit [36]. A comparison between methods also supports the 1 M nitric acid method as the best predictor for total Pb [21]. The pXRF provides an accurate express survey of urban ecosystems in field conditions [37]. In this study, the Berry–Mielke Universal ℜ agreements are notably strong among the methods.

### 4.2. Children’s Blood Pb in St. John’s

The results for all soil analysis methods indicate that the soil Pb content is too high and unsafe for the health and welfare of St. John’s children. Given the laboratory results precautionary advisories and soil Pb remediation recommendations to reduce high-Pb soils at urban gardens and children’s play areas need to be issued [38]. The expectation was that the children of St. John’s would exhibit high blood Pb. The next logical step was taken to conduct a blood Pb survey of the children of St. John’s. 

As a public health issue, the survey was funded by Health Canada to conduct blood Pb sampling of the children of St. John’s NL [10,39]. The Health Canada surveillance included 257 children < 6 years old. The children’s blood Pb samples were drawn during the summer and fall months when exposures were expected to be highest. The resulting survey found a median blood Pb of 1.04 µg/dL; the 95th percentile was 2.71 µg/dL, and one child measured 7.4 µg/dL and above the ≥ 5 µg/dL blood Pb reference value [10]. The blood Pb survey does not show signs of high-Pb exposure for children living in the small city of St. John’s, NL. 

The lack of an expected association between children’s blood Pb and soil Pb in St. John’s NL is a critical issue that requires further investigation to obtain perspectives on urban processes that affect children’s blood Pb. If St. John’s children’s blood Pb results are not associated with soil Pb, then what processes are diminishing exposure risk from the Pb contaminated soil of St. John’s?

The simplest possibility is that the first step in sample preparation, the dry heating and sieving of soil samples through screens sorting the samples to <63 micrometer particles might influence the results. This step may concentrate the smallest and most toxic particles. One speculation is that the coal contaminated soils of St. John’s are coalesced into larger particles that do not adhere to the hands of children and that, furthermore, larger particles are not prone to being resuspended into the air. This speculation can be experimentally investigated. Other known urban processes related to children’s Pb exposure must also be considered. 

### 4.3. City Size, Lead Particle Pollution, and Exposure Risk Factors

Perspective about processes that cause Pb accumulation have been garnered from various urban soil Pb research studies. For example, in the 20th century, Pb sources included ~6 million tonnes of anthropogenic Pb dust exhausted into the air by motor vehicles [40]. Urban Pb pollution became a global problem because the major additive, tetraethyl lead (TEL), was universally added to petrol in cities throughout the world [41]. The total tonnages of Pb along with quantities by particle size from exhaust have been estimated for 90 US cities [40]. The smallest particles (<2.5 µm, and especially <1 µm ultrafine size) are particularly toxic because ultrafine particles are inhaled and pass into the blood stream. Soil is not ordinarily treated an active reservoir of urban Pb dust exposure [12,13]. However, research in the UK demonstrated a dynamic linkage between Pb isotopes of 20th Century leaded petrol and current releases of Pb isotopes into the atmosphere of London [22]. St. John’s is a small city, and Pb sources and particle sizes matter.

The traffic flows and urban congestion patterns corresponds to quantity of accumulated Pb as shown by the findings from 90 U.S. cities [40]. For more specific examples, in Baltimore, Maryland, Pb in garden soils in the inner-city brick building communities had higher accumulated Pb quantities than garden soils in outlying communities where lead-painted buildings were common [5]. Other studies demonstrate the role of city size on soil Pb. For example, in the Australian cities of Sydney, Melbourne, and Brisbane, Australia, the amount of soil Pb is directly associated with city size [42] (Table 2). The same city size and soil Pb relationship was observed in the U.S. cities of Minnesota (Minneapolis, St. Paul, St. Cloud, and Rochester) and the cities of Louisiana (New Orleans, Baton Rouge, Monroe, Lafayette, and Natchitoches), which show a strong association between city size and soil Pb content [14,43]. Garden soils of New York City have been assessed and show a strong geospatial pattern of contamination [44].

Several studies show strong associations between soil Pb and blood Pb. Examples are available for the following cities: Syracuse, NY; Minneapolis and St. Paul, MN; New Orleans, LA; [32,45,46,47,48]. In each of these larger cities, strong associations were identified between soil Pb and children’s blood Pb, highest soil Pb and blood Pb results near the city center and lowest results away from city center. Furthermore, in New Orleans when the use of leaded petrol was curtailed, the Pb loading of soil decreased over ~15 years, and blood Pb also decreased [14]. The same temporal pattern of decreasing soil Pb and blood Pb was similarly noted for the Detroit Tri-County area of Michigan [49].

At the national scale in the U.S., when leaded petrol use declined, children’s exposure to Pb aerosols and their blood Pb markedly decreased [50]. In China, after TEL was banned in petrol in 2000, a similar decreasing trend occurred among cities [51]. The ban of Pb in petrol has had a profound effect on the soil Pb and blood Pb of children living in urban environments, and the larger the city, the more intense the effect. 

St. John’s, NL, is a small city (population < 120,000), approximately half the population of the smallest city, Laredo, TX, with estimates of accumulated Pb from petrol [40]. The soils of St. John’s have a high-Pb content as determined by four analytical methods. The soil was assumed to be contaminated by high-Pb coal ash and paint chips. However, the remarkably low blood Pb of St. John’s children is not consistent with the existence of high soil Pb. More investigation is needed on soil contaminated by high-Pb coal ash in particular.

Multiple studies noted that the patterns of soil Pb and blood Pb are associated with city size and locations of communities within urban areas [14,32,45,46,47,48,49,51]. St. John’s is a small city, and that circumstance limited the 20th century accumulation of Pb particles. The outcome is an amazingly reduced blood Pb among children living in small cities in the 21st century. 

## 5. Conclusions

High-Pb coal was used for heating in St. John’s, NL, and the Pb contaminated ashes were deposited on residential soils. In addition, wooden clapboard houses were painted with leaded paint. This environmental research compared four common Pb analysis methods for agreement on the same Pb-contaminated soil samples (N = 96) collected from the small town of St. John’s, NL. Primary prevention requires monitoring methods to avert Pb exposure in the first place. This study contributes agreement analysis of methods assessing Pb in the environment to avert the Pb exposure of children.

Out of the four methods, two methods, the 1 M nitric acid extraction and the pXRF method, were best in terms of high agreement, labor requirements, and convenience. Given the minimal sample preparation and high agreement with other methods, the cost for an pXRF is in line with its usefulness as a primary prevention tool in children’s exposure to soil Pb.

This study emphasizes that agreement is not the same as correlation. The Berry–Mielke Universal ℜ coefficient of agreement shows that all four methods are in strong agreement, and this finding has far-reaching implications. It means that the results from one method compared with the result of another method produces the same opinion and decision about the degree of Pb hazard posed by St. John’s contaminated soil. The strong agreement between the methods indicates that the tested St. John’s soils are typically too Pb-contaminated for children. 

However, a public health reality check funded by Health Canada for a blood Pb survey was conducted on young children (N = 257) living in St. John’s. The survey revealed an exceptionally low blood Pb for the children of St. John’s. 

The quantity of leaded petrol used during the 20th century must be acknowledged as a principal factor in urban soil Pb accumulation. Many urban soil Pb studies underscore the idea that city size and location within the city are factors in the 20th century dynamic processes that changed soil Pb and that continues to affect children’s blood Pb exposure in large and small cities. Small cities had minor traffic congestion and limited pollution from combustion particles from leaded petrol. Despite the elevated Pb levels in soil from presumably coal ash and leaded paint, children living in St. John’s, NL, have low blood Pb, and this is characteristic of children living in small cities.

## Figures and Tables

**Figure 1 ijerph-18-09863-f001:**
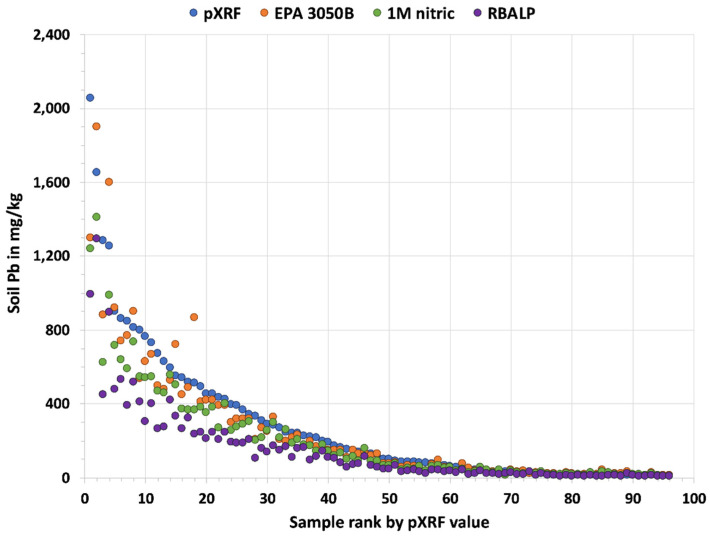
The graph shows the soil Pb results for each soil sample (N = 96) ranked by pXRF value for the results from each of the analytical methods.

**Table 1 ijerph-18-09863-t001:** Percentiles and Universal ℜ agreement coefficients with *p*-values for the 4 soil Pb analysis methods compared in this study. Medians are given as bold font.

	Method				Comparison	ℜ	*p*-Value
Pb (mg/kg)	pXRF	EPA3050B	1 M Nitric	RBALP			
N	96	96	96	96	pXRF vs. EPA 3050B	0.832	<0.0001
min	15	13	10	7	pXRF vs. 1 M nitric	0.765	<0.0001
5%	15	16	13	10	pXRF vs. RBALP	0.552	<0.0001
10%	15	21	17	11	EPA 3050B vs. 1 M nitric	0.833	<0.0001
25%	26	31	28	21	EPA 3050B vs. RBALP	0.609	<0.0001
**50%**	**113**	**103**	**84**	**59**	1 M nitric vs. RBALP	0.741	<0.0001
75%	396	328	297	203	All four methods	0.727	<0.0001
90%	777	726	551	405			
95%	953	903	718	522			
max	2058	1900	1410	1295			

**Table 2 ijerph-18-09863-t002:** Equipment needs, treatment of soil during preparation, stomach pH and temperature issues, safety, and productivity factors that should be considered when undertaking extensive urban soil survey monitoring projects.

	EPA 3050B	pXRF	1 M nitric	RBALP
Equipment needs	Microwave apparatus, Teflon vessels, ICP-MS	Handheld XRF instrument	Shaker, centrifuge, AAS or ICP-OES	Special apparatus, ICP-OES
Soil preparation	Heat drying, grinding, sieving	None to drying for reducing moisture (over 20%)	Indoor air drying and sieving	Several steps required to maintain pH
Mimic stomach pH	No	Not applicable	No *	Assumed to be yes
Physiological temperature	Operates at high pressures and temperatures	Not applicable	No, room temperature *	Yes
Safety	Special venting, concentrated acid	Radiation hazard training	Safe in open laboratory, dilute acid	Safe chemicals
Sample throughput	Low	High	High	Low

* The relevance of the method to human health in larger cities is demonstrated by the strong association, across a spectrum of communities, between soil Pb and children’s blood Pb [14,32].

## Data Availability

Appendix A accompany this manuscript containing complete lists of Pb data for each of the methods for St. John’s soil plus a supplemental figure of pairwise LAD regressions of the analysis results.

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
