# Peer review of "Agreement ℜ of Four Analytical Methods Applied to Pb in Soils from the Small City of St. John’s, Newfoundland, Canada"

_ijerph, 2021, doi:10.3390/ijerph18189863_

Round 1
Reviewer 1 Report
Much improved although I have one issue with the manuscript before acceptance for publication.
Lines 330-331 are speculation and should be framed as a hypothesis to be tested. This could be supported by results from other cities containing children with reduced blood Pb, lower than would be expected based on soil Pb levels. Also change the last phase of conclusions, line 358.
Author Response
Fundamentally, the statement on lines 330-331 is not speculation because research has long established that children living in small cities have lower blood Pb (Minnesota cities and Detroit tri-county cities (cites 14 and 40) . We state clearly on lines 278-284 that the process in St. John's is speculative, and suggest an experiment on particle sizes of coal ash that may provide insight into the findings from the finely ground samples supplied from St. John's for this comparison study. Research also shows the relationship between soil Pb and blood Pb published in (EHP 1997, Mielke et al. 105(9):950-954, Table 6, and #14 PNAS (Table 1).
Reviewer 2 Report
Dear Authors,
your manuscript is original, interesting and deserve publication.
I would suggest to draw a correlation between low Pb in blood in spite of high contamination and climatic factors. All the examples quoted in the discussion refer to continental towns at lower latitudes than St. John (i.e. higher average T, and enhanced kinetics of bioreactions).Snow cover for several months inhibits the contact of children with soils (to be compared with Michigan case). Also dominant winds affect this area and potentially disperse the aerosol.
Kind regards,
Laura Gaggero
Author Response
Thank you for your encouraging words regarding your interest and your suggestion that this agreement study deserves publication.
We introduce agreement as the most appropriate statistical test. Correlation is not the same as agreement.
Other cities have been included that are higher latitude cities such as Minnesota, Michigan, NYC, Australia, Syracuse NY, and China (citation #s 34,37,40,44,47, 51). These cities represent a wide range of lower to higher latitudes.
We focus on urban soils. Snow cover at higher latitudes does influence potential intake in winter, but given that the blood Pb survey was conducted in the snow-free months, soil Pb should have had a stronger influence on blood Pb. We speculate that the Pb in finely ground samples provided from St. John's were more easily extracted than would be the case for the original soil samples. We suggest that experiments needs to be conducted to determine that possibility.
Thank you for your comments.
Reviewer 3 Report
Comments to the Author
1. Reorganize the line from 67 to 70.
2.line 184-185:“As shown in Table 1 the differences of percentiles show that all results 184 are above soil Pb levels recognized as safe.”What is the safety level?
3.line 328-329:” Multiple studies noted that the patterns of soil Pb and blood Pb are associated with city 328 size and locations of communities within urban areas.” It need a lot convincing references.
4.Can the conclusions of this paper be applied to other regions? Such as the city soil in Africa, Asia and so on.
Author Response
Thank you for your comments and suggestions.
- We reorganized lines 67-70. "We sought to compare the results of the methods to find out how they agree with each other. The comparison investigation took on special meaning because after decades of studies with the Chaney-Mielke method (developed in the mid-1970’s)."
- A safe soil Pb exposure for children is not well established. The research by CA regulators have a safe soil Pb of 80ppm, but it was established when blood Pb was 10 micrograms per dL. Now there is no known safe Pb exposure. In New Orleans, the safe level appears to be around 40 ppm for half the population (assuming that soil Pb is the only source of exposure). The answer to this question has not been resolved to protect children. It has been resolved on the basis of financial interests of the lead industry.
- This is not a review article on the association between soil Pb and children's blood Pb. We have provided excellent peer reviewed research articles from reliable sources. Please read the articles cited.
- Definitely, thes conclusions are the subject of studies in a wide range of countries from low income to high income nations. Citations are from the US, Australia, Europe, and China and provide insight into soil Pb and in some cases childhood blood Pb as well. Good work has been done in Africa.
Reviewer 4 Report
The article describes the studies on procedures of determination of lead in soil samples. However, the article is well constructed, but a few of problems raises doubts and should be clarified. My recommendation is the major revision of the text. I hope that proposed changes improve the value of the text.
General comments:
The novelty of the described work should be better underlined (i) the procedures of lead determination are well-known and the difference of present work should be pointed (ii) scientific problem seems to be rather local, but the studies may be interesting for wider readership.
The concept of analytical method agreement should be better defined. Each of the indicated methods determines a different feature of the tested samples and authors should explain the purpose of their research.
For results interpretations the main important point is the quality of data obtained in analysis.
The information about instrumental parameters and detailed description of sample preparation procedure should be added.
The information about analytical procedures (completed if possible by information about basic metrological parameters and estimation of the uncertainty) should be added. The lack of confirmation of the results in the CRM analysis makes it impossible to apply it in practice.
Detailed question (clarification of the following issues is necessary to show the proper conduct of analytical measurements):
XRF instrument was calibration and the strategy of control of quality of measurements should be described.
Which Pb isotopes were determined in ICP-MS, how the isobaric interferences has been corrected?
The background correction and the correction of optical interferences in ICP-OES should be described.
Authors indicate the same detection limit 2 mg/kg for two different procedures of sample preparation with different factors solid sample/solution (50 v. 100). How the detection limit for ICP-OES measurements was calculated?
Author Response
The article describes the studies on procedures of determination of lead in soil samples. However, the article is well constructed, but a few of problems raises doubts and should be clarified. My recommendation is the major revision of the text. I hope that proposed changes improve the value of the text.
Thank you for your kind comments about the construction of the manuscript. It is not apparent to the authors that this manuscript was a complete revision of a previous submission. I am responding under pressure from a major destructive Hurricane that is expected to make landfall in 6 hours. The power will fail when Hurricane Ida makes landfall.
General comments:
The novelty of the described work should be better underlined (i) the procedures of lead determination are well-known and the difference of present work should be pointed (ii) scientific problem seems to be rather local, but the studies may be interesting for wider readership.
The concept of analytical method agreement should be better defined. Each of the indicated methods determines a different feature of the tested samples and authors should explain the purpose of their research.
For results interpretations the main important point is the quality of data obtained in analysis.
Of course, the quality of the data is essential. We do not agree as the reviewer seems to suggest that the analysis done for this study lacks quality. The team of researchers has a long record of high-quality research and analysis as recognized by many reviewers including citation #14.
The information about instrumental parameters and detailed description of sample preparation procedure should be added.
Each of the methods includes citations that provide the necessary descriptions and direction for conducting each of the extraction methods.
The information about analytical procedures (completed if possible by information about basic metrological parameters and estimation of the uncertainty) should be added. The lack of confirmation of the results in the CRM analysis makes it impossible to apply it in practice.
We do not agree with the reviewer about the value of the additional information suggested in these general comments. The idea that lack of CRM analysis makes this impossible to apply in practice does not match the experience of many researchers using the 4 methods described in this manuscript. Furthermore the fundamental contribution of our study is that agreement is not the same as correlation. This study uses the results from 4 methods to convey the value and importance of using agreement for comparing results.
We thank the reviewer for comments about our manuscript. We do not believe the reviewer is aware that this manuscript is a completely revised and rewritten manuscript based on previous reviews.
Detailed question (clarification of the following issues is necessary to show the proper conduct of analytical measurements):
1. XRF instrument was calibration and the strategy of control of quality of measurements should be described. Analysis was conducted with a Niton XL2 handheld XRF for 30 seconds on < 63 µm particle size soil samples. The detection limit is 15 mg/kg [7,8,24]. Calibration was done internally and then checked using a NIST standard and two laboratory soil standards used for all analytical work.
2. Which Pb isotopes were determined in ICP-MS, how the isobaric interferences has been corrected? We cited a peer reviewed Proceedings of the National Academy article (citation # 22) regarding the Pb isotope study. It is not our research study and the details are given in the cited manuscript.
3. The background correction and the correction of optical interferences in ICP-OES should be described.
The operating procedure includes correction of optical interferences and background correction. The research team is experienced and fully aware of the necessity of operating the Spectro ICP according to specified operating procedures.
4. Authors indicate the same detection limit 2 mg/kg for two different procedures of sample preparation with different factors solid sample/solution (50 v. 100). How the detection limit for ICP-OES measurements was calculated?
The Spectro ICP-OES limit of detection was calculated according to standard operating procedures for the analytical instrument. The Team has years of experience and pays careful attention to the necessary requirements of metal analysis.
Round 2
Reviewer 4 Report
Due to the fact that the authors did not introduce most of the suggested changes, I still think that the revision of manuscript is necessary.
General comments
The reviewer does not know the history of the manuscript preparation. The Editor's decision was to request a review, in line with the journal's publication policy. Whether the article has been reviewed earlier does not affect its assessment. The authors repeatedly emphasize their experience. The reviewer, after nearly 30 years of working with spectrometric techniques, indicates elements of the research description that could be improved.The reviewer does not question the team's qualifications, as the authors suggest. However, it indicates fragments of the research description that should be explained.
The authors indicate that they included literature references in the text, allowing for the repetition of the research. However, the text still lacks information on instrumental parameters important for obtaining results, e.g. for ICP-MS device model, plasma flow, nebulizer flow, RF power etc.
In analytical practice, the use of CRM (or the standard addition method) is the most important element of traceability of results. In publishing practice, studies without proving the accuracy of the results cannot be accepted in most journals.
Detailed comments
Selection of Pb isotopes is very important in ICP-MS measurements. The question did not concern the determination of the content of various lead isotopes in the environment, but only the selection of the monitored isotope or isotopes in ICP-MS. Moreover, the authors did not define the method of eliminating isobaric interferences, an issue important in the analysis of ICP-MS.
The reviewer does not question the authors' experience, but only asks them to share this experience with the readers and to supplement the information on instrumental procedures.
Once again, the reviewer does not question the authors' experience. However, different sample preparation changes the limits of detection for the samples!
Author Response
IJERPH-1360267; Responses to reviewer 4 on the second draft.
We answer the last two comments first.
In analytical practice, the use of CRM (or the standard addition method) is the most important element of traceability of results.
In our analysis we use National Institute of Standards and Technology (NIST) traceable standards for ICP-AES calibration verification. This is part of the SOP for the ICP-AES. The NIST standards are purchased and certified by the SPEX. We have several sets of purchased standards from different chemical suppliers to double check the ICP calibration. We also have other certified standards from the Netherlands that we use as a check for the operation of the ICP and its calibration.
In publishing practice, studies without proving the accuracy of the results cannot be accepted in most journals.
In publication practice, this statement is simply untrue. What journals is the reviewer referring to? The usual journals recognize accuracy of results along with the track record of the researchers and quality of the manuscript. Furthermore, attention is paid to the content of the study, ideas expressed, and the supporting references in the manuscript. Proving accuracy is not an undertaking required in ordinary publications. Proving accuracy is a massive project that is described by the US EPA. The following reports provide details about the methods that can be referred to in a manuscript for details about the method:
- See reference [3]. US EPA, 1996. Method 3050B: Acid Digestion of Sediments, Sludges, and Soils, Revision 2. Washington, DC. US EPA, 2006. https://www.epa.gov/sites/default/files/2015-06/documents/epa-3050b.pdf
- reference [6]. US EPA, 2013. Method 1340 in vitro bioaccessibility assay for lead in soil. SW-846 hazardous waste test methods. https://www.epa.gov/sites/default/files/2017-03/documents/method_1340_update_vi_final_3-22-17.pdf
- and reference [26]. Drexler, J.W., Brattin, W.J., 2007. Procedure for Estimation of Lead Relative Bioavailability: With Validation. Hum. Ecol. Risk Assess. An Int. J. 13, 383–401. https://doi.org/10.1080/10807030701226350
The authors indicate that they included literature references in the text, allowing for the repetition of the research. However, the text still lacks information on instrumental parameters important for obtaining results, e.g., for ICP-MS device model, plasma flow, nebulizer flow, RF power etc.
The original samples were measured for Lead NL by Maxxam Analytics in Bedford, Nova Scotia. Maxxam Analytics in Bedford Nov Scotia is an accredited ISO 17025 certified laboratory. The certified lab has an ICP-MS, but it does not supply the information the reviewer is asking for. The certified laboratory does sample analysis according to the US EPA SW-846 Method 3050B [3]. The parameters for the ICP-MS at Maxxam Analytics in Bedford, Nova Scotia are not available. The laboratory is an accredited ISO 17025 certified laboratory.
A description of the ICP-AES instrument used in New Orleans for analysis is a CIROS CCD (2003) with Axial view of the Argon plasma. Spectro CIROS CCD S/N: 4N/0145 and manufactured by SPECTRO Analytical Instruments GmbH, Kleve, Germany. This is an older but reliable instrument from German engineers. The setup is from a series of instrument guides that are laid out in the computer instructions for use of the instrument. This is a typical protocol for these instruments. To avoid operator error, if the instrument is not set up appropriately, then for the sake of safety it does not run. The instrument is regularly cleaned and checked by Spectro engineers to keep it operating at peak efficiency.
All instruments have different setup protocols and guides for operation.
Tulane ICP-AES SOP. The 1 M nitric acid soil sample extracts follows the protocol provided in the manuscript [25]. National Institute of Standards and Technology (NIST) traceable standards were used for ICP-AES calibration verification. Duplicate soil samples at a rate of 1 per 20 samples were prepared and analyzed. In-house reference soil samples were included during analysis, and the SPb results for these NIST and inhouse samples were consistent in the study. If the NIST traceable standards should fail on a given run, then all soil samples in the given run are reanalyzed. Furthermore, all samples are archived at Tulane University, so at any time if there is a question they can be reanalyzed.
The RBALP was performed in our laboratory following the guidelines specified in Drexler & Brattin [26] and the US EPA METHOD 1340 [6]. The Tulane ICPS-AES SOP was used to measure the samples.
Selection of Pb isotopes is very important in ICP-MS measurements. The question did not concern the determination of the content of various lead isotopes in the environment, but only the selection of the monitored isotope or isotopes in ICP-MS. Moreover, the authors did not define the method of eliminating isobaric interferences, an issue important in the analysis of ICP-MS.
There are two references to isotopes. The Maxxam Analytics in Bedford Nov Scotia is an accredited ISO 17025 certified laboratory, and it has an ICP-MS instrument. Ask them. It is not our analysis.
The other paper referring to isotopes is the London study. This is not our study. It was conducted by researchers at Imperial College in London and the study was peer-reviewed by the editors of the Proceeding of the National Academy of Sciences.
We urge the reviewer to carefully read the cited reference to the London Pb isotope study and ask your questions of the authors of the ICP-MS research. [22]. Resongles et al. (2021). Strong evidence for the continued contribution of lead deposited during the 20th century to the atmospheric environment in London of today. PNAS https://doi.org/10.1073/pnas.2102791118.
We included the study, as is normal practice, because it has relevance to our study of Pb in urban environments.
The reviewer does not question the authors' experience, but only asks them to share this experience with the readers and to supplement the information on instrumental procedures.
Once again, the reviewer does not question the authors' experience. However, different sample preparation changes the limits of detection for the samples!
The issue about limits of detection (3 standards of deviations of above the mean) of the samples is irrelevant to the soil samples that were analyzed in this study. At no time did we encountered a sample that was even close to 2 mg/kg, nor did we encounter a LQD (limit of the quantitative detection of even 5) i.e., 10 mg/kg. This comment does not fit with the data we report on in the manuscript.
This manuscript is a resubmission of an earlier submission. The following is a list of the peer review reports and author responses from that submission.
Round 1
Reviewer 1 Report
General evaluation
The title of the paper corresponds weakly with the paper’s content. The manuscript lacks a clear research objective and, hence, lacks focus. Results and conclusions are to some extent inconsistent. The presented data do not constitute sufficient new information to justify an international scientific publication.
Comments to the chapters
Abstract
Sixty percent of the Abstract is introductory literature review.
The conclusions that “High Pb coal ash contaminated soil is unequal to vehicle Pb exhaust contaminated soil” (line 23) and “Particle sizes of the source of Pb contamination and city size are risk factors in evaluating the consequences of soil Pb on children’s exposure” (lines 23–25) are not based on the presented data.
Introduction
There is a disconnect between the introduction and the stated research objective. The chapter introduces the readers to the elevated levels of Pb in soils in St. John’s, NL, Canada, whereas the research objective is “to apply four of the best known Pb extraction and analytical methods to the same high Pb soil samples”. If the purpose of the study is to compare the analytical methods, the introduction should outline previous studies on Pb analysis and explain why there is a need to perform a comparative study on the extraction and analysis of Pb. Some explanation is given after the objective statement, but this is not sufficient. The authors should describe existing methods and explain how they correlate with Pb levels in human blood. Furthermore, strengths and weaknesses linked to the available methods should be outlined. The need to compare the selected four methods (the current research objective) should be clearly stated.
The purpose of all research projects should be to obtain new knowledge. The current objective “to apply four of the best known Pb extraction and analytical methods to the same high Pb soil samples” is not a proper research objective. To simply “apply ... methods” cannot be a scientific research objective.
Materials and Methods
The chapter is detailed and well written. But while reading it, this reviewer was still wondering why this work was undertaken. No satisfying justifiction seems to be given.
The authors should go through the manuscript and reevaluate capitalizations. For instance, “Low-Density Polyethylene”, “Value” (as in “P-Value”), “Nitric” (nitric acid?) and “Coupled Plasma-Mass Spectrometry” are common nouns and should, therefore, not be capitalized.
Results
Comparing results to those reported in the literature (lines 149–150 and 167–169) constitutes discussion of results. Hence, these sentences should be moved to the Discussion chapter.
The term “1M Nitric” (line 150 and other places) should be ‘1M nitric acid’.
Discussion
Most of this chapter is literature review, some of which would be suitable in the introduction to the paper. The text is for the most part not a discussion of current results, but a discussion of background information as well as a general discussion of heavy metals in the environment.
The data for children’s Pb levels in their blood was taken from Health Canada published in 2013. Hence, these data cannot be presented as ‘results’ in this paper. The data can only be used to obtain new information based on additional data.
The information in lines 191–199 does not constitute discussion of results. The text is a literature review that technically belongs in the Introduction chapter. The authors should keep the chapters more thematically focused.
Conclusion
The conclusions that “Generally, soil is an underappreciated source of Pb exposure” (line 310) and “St. John’s soil is too Pb contaminated for children and precaution is advised” (lines 330–331) appear to be inconsistent with the previous statement that “St. John’s blood Pb results demonstrate that the children have remarkably low Pb exposures” (lines 253–254). It appears that the authors base their conclusions on the literature rather than their own data.
Also, much of the Conclusion chapter (lines 312–322) is simply a summary of methods used. Such information should not be presented as conclusions.
Reviewer 2 Report
The manuscript is about a relevant soil pollutant, and about comparing four different techniques to determine Pb in soil. Although the methods used are excellent, the manuscript is not written in a very scientific style: it has many self-citations and only citations to North American studies and problems. Although it is about St Johns NL, Canada, it is relevant also for other polluted cities in the world.
Only a few aspects need to be improved: the choice of the soil samples is not discussed. A statistical approach is preferred in science, but the authors do not discuss the soil sampling at all. The second problem is that the whole manuscript is filled with self-references, North American references and descriptions of North American problems. That is not appropriate for the problem which is international.
Detailed comments
46-50. This part of the text but also other parts of the text only contain references to North American research and North American polluted locations. That is not appropriate as it gives an incorrect impression of the problem.
56 “The purpose of this study is to apply four of the best known Pb extraction and analytical methods to the same high Pb soil samples”. & 80 “Table 1 lists soil Pb extraction methods which have been described in the literature””.
Selecting four methods out of four methods is not the same as chosing the “four best known” methods or listing methods which have bene described in the literature”. For example a method that includes the use of isotopes makes it possible to determine the source of contamination (Walraven https://doi.org/10.1016/j.apgeochem.2013.07.015) and the 0.43 M HNO3 extraction has been related to human oral bioaccessibility (Rodrigues et al 2018 https://doi.org/10.1016/j.scitotenv.2018.04.063). Please be more clear about the reasons for chosing these four methods.
67 Please add the soil depth of the samples taken to determine the Canadian background Pb concentration. Is glacial till a selection of the dataset, excluding peat soils?
74 How were the soil sampling locations chosen? Was there a statistical approach, have locations been chosen on the basis of play grounds (source of exposure), were locations chosen on the basis of previous research etc etc? Please give a table with the type of locations, number of samples for garden, playgrounds, roads, forest, agricultural soil etc. In line 293 you state ”the study was conducted on the same high Pb contaminated soil samples”. That suggests that you took a selection from a previous study.
80 XRF is not an “extraction method”.
80-106. You add the detection limit to all four methods. In practice the repeatability is often larger problem. Can you add the repeatability as determined on ring test sample?
182 “residual coal ash”. It is not clear for the reader how this waste material is used. Was it used in the garden, roads, paths?
200-221. Thee arguments sound rather academic as only one of the methods is performed in commercial laboratories, is known in ring tests, and has a well-known price.
241-243 “The children’s blood Pb were most strongly correlated in a univariate analysis with soil Pb concentrations from play areas” this seems to be in contradiction to: 249 “The lack of association between soil Pb and blood”. As a reader I do not understand this.
251-255. Did you have soil samples from play grounds? Did you have soil samples that were analysed previously, and how did these compare?
262 Are there other types of industry of product that use or contain Pb? Painted rooftiles are a important source of lead in old cities in Europe, and various metallurgical industries. Please just address if all sources of Pb have been looked into.
Chapter 4.4 & Table 2 You discuss pXRF, but all the soil samples were sieved to < 63 um. That does not sound like a simple method that can be used in the field.
291 “underappreciated”? Why do you conclude this, and why do you start with this statement which is just an impression? Pleas skip this.
308-310 I think you cannot conclude this in general as you did not include polluted samples from other sources of pollution.
Reviewer 3 Report
This manuscript has several shortcomings, broadly falling in the categories of writing and of the findings of the work.
Overall, the writing is poor: very repetitious, often unclear, poorly organized. It is also unnecessarily too technical, the perfect example being section 2.4: it is perfectly unintelligible by the common mortal. Maybe the authors will feel that the text is very clear, but I was often lost, so a strong effort of re-writing the text must be undertaken.
There is also too much irrelevant text that has nothing to do in this paper, such as discussion the health implications of high Pb in the soil, the origin of Pb, or remediation. For instance, sections 4.3. and 4.4 can safely be removed: they are purely speculative, are not derived from the data presented, and have nothing to do with the purpose of the paper. All these topics may be valuable and important, but this paper is supposed to deal with the Pb analytical methodology; all the rest is fluff.
The second set of problems is the findings. The authors conclude that all the methods examined are valuable (line 307), which is definitely not supported by the data (table 2). The median obtained by pXRF is 113 ppm, but the RBALP method returns 59 ppm: obviously they are not both correct. Even if we exclude the RBALP method, there is still a massive difference between the median obtained by pXRF and that by 1M Nitric (84 ppm; table 2). The manuscript fails to clearly comment on these differences and indicate which of these methods provides the best results (section 4.1 is unclear at best). In other words, this manuscript does not deliver on its central purpose, which is comparison between the methods.
Overall, this is a weak manuscript. It can be saved by completely re-interpreting the data and re-writing the text, but in its present state it is unfit for publication.
Reviewer 4 Report
Comments and Suggestions for Authors:
The paper titled “Effects of Comparison of four measurement methods on Pb contaminated soils from the small city of St. John’s NL, Canada” deals with four of the best soil Pb analysis methods to compare their efficacy on St. John’s contaminated soils. The purpose of this study is to apply four of the best known Pb extraction and analytical methods to the same high Pb soil samples.
In general, the manuscript is well structured and comprehensive. The subject of the present research is interesting. What the authors want to say in the present study is well said and stated. The research is properly organized and conducted. After all, in the authors there are included significant researchers concerning this subject. Hence, their contribution is apparent throughout the whole manuscript.
However, it would be very auspicable if at the end of such an interesting and informative introduction, the authors explained more clearly the novelty aspects of the present research. Please refer if any similar studies have been previously carried out. If so, then please state why this study excels compared to other similar. In my opinion, this would make the manuscript more interesting to the reader.
Furthermore, throughout a manuscript concerning such an interesting subject, in some cases it would be better if literature would be enriched. For example, in the first paragraph of the “Introduction” part (rows 29-39) only one reference is referred. It is just cited 3 times throughout the whole paragraph. In my opinion, at this point the authors could add some more references than just [1].
Regarding the English language style, although I do not feel qualified to judge about it, it must be said that it is very comprehensive. Minor corrections like the following are required:
Row 69:
“higher concentrations” probably would be better expressed as “elevated concentrations”
Row 82:
“The other three methods” probably would be better expressed as “The rest three methods”
For these reasons, minor revision should be carried out before publication, while some comments and suggestions for the authors are listed below.
Row 127:
Please alter “pb” with “Pb”
Rows 139+141:
At this point, the decimals are divided by a comma, while in the rest of the numbers by a dot. Please choose a uniform manner throughout the whole manuscript.
Rows 107-126 (2.4. Statistical analysis):
It would be interesting if the authors would explain the reason why they chose to calculate the Berry-Mielke Universal R coefficient, instead of Pearson (r) or Spearman (ρ) correlation factors which are more commonly used in similar studies.
Moreover, at some points of the manuscript (rows 106, 139, 140, 141) the form “mg Pb/Kg” should probably be replaced with the right one “mg/kg Pb”. Please consider and revise.
Finally, maybe the authors should consider the possibility of the graphical presentation of the results instead of just mentioning their variation (as in rows 139-141). There is a lot of numerical information obtained by the research, but they are mostly referred in tables and in the text. It would be much better for the presentation of the results if they were demonstrated in diagrams. That way, the manuscript would become much more comprehensive to the reader.
Reviewer 5 Report
This is a potentially interesting study. Four methods for analysing Pb were compared to each other, while results for children's Pb levels in blood were also presented. What I expected was an assessment of which method(s) was/were best for evaluating the risk to children. However, this is missing. XRF is a well-established screening method for Pb, while the low blood levels in St John’s children was published several years ago.
Other improvements are needed.
The English needs improving thoughout the paper.
The paper structure needs improving (move random pieces of information to their own or a more correct section, e.g. lines 98-99).
Where is table 3?
Remove unsubstantiated statements, e.g. “soil is an underappreciated source of Pb exposure” when soil is well documented in the literature as both a sink and source of Pb. Nothing in this paper suggests otherwise.
Reviewer 6 Report
In my opinion, the manuscript needs careful reading to correct grammar and typos in the text.
Examples of sentences needing to de adequately revised:
Lines 110 to 111 – “We use the data-dependent, non-parametric Multi-Response Permutation Procedure (MRPP) analysis uses all available, non-transformed, raw data [27–29].”
Line 257 – “A major source of environment Pb emissions that accumulated in soil from motor vehicles that previously used leaded gasoline in the 1920s through the 1980s [43].”
Table 1 in Section 2.3. – is it missing? Or it refers to table 1 that is presented in the manuscript?
Two “Table 2” are provided